# Assessment of Gastrointestinal Adverse Effects during the First Six Months of “Biktarvy” Antiretroviral Therapy: Age-Related Patterns and Their Relation with Changes of 5 kg Weight Loss/Gain in the Initial Treatment Period

**DOI:** 10.3390/diseases12010003

**Published:** 2023-12-21

**Authors:** Madalina-Ianca Suba, Simona-Alina Abu-Awwad, Ovidiu Rosca, Mirela-Mădălina Turaiche, Bogdan Hogea, Ahmed Abu-Awwad, Razvan Nitu, Voichita Elena Lazureanu

**Affiliations:** 1Doctoral School, “Victor Babes” University of Medicine and Pharmacy, Eftimie Murgu Square, No. 2, 300041 Timisoara, Romania; madalina.suba@umft.ro; 2Dr. Victor Babeș Infectious Diseases and Pneumophthisiology Hospital Timisoara, 300310 Timisoara, Romania; 3“Pius Brinzeu” Emergency Clinical County Hospital, Bld Liviu Rebreanu, No. 156, 300723 Timisoara, Romania; hogea.bogdan@umft.ro (B.H.); ahm.abuawwad@umft.ro (A.A.-A.); nitu.dumitru@umft.ro (R.N.); 4Department XIII, Discipline of Infectious Diseases, “Victor Babes” University of Medicine and Pharmacy Timisoara, Eftimie Murgu Square 2, 300041 Timisoara, Romania; ovidiu.rosca@umft.ro (O.R.); lazureanu.voichita@umft.ro (V.E.L.); 5Methodological and Infectious Diseases Research Center, Department of Infectious Diseases, “Victor Babes” University of Medicine and Pharmacy Timisoara, Eftimie Murgu Square 2, 300041 Timisoara, Romania; paliu.mirela@umft.ro; 6Department XV, Discipline of Orthopedics-Traumatology, “Victor Babes” University of Medicine and Pharmacy, Eftimie Murgu Square, No. 2, 300041 Timisoara, Romania; 7Profesor Universitar Doctor Teodor Șora Research Centre, “Victor Babes” University of Medicine and Pharmacy, Eftimie Murgu Square, No. 2, 300041 Timisoara, Romania; 8Department of Obstetrics and Gynecology, Faculty of Medicine, “Victor Babes” University of Medicine and Pharmacy, 300041 Timisoara, Romania

**Keywords:** bictegravir, emtricitabine, tenofovir alafenamide, BIKTARVY^®^, antiretroviral therapy, HIV, gastrointestinal adverse effects, age-related patterns, weight loss, treatment outcomes

## Abstract

Background: The battle against HIV has led to the development of antiretroviral therapy (ART), including BIKTARVY^®^, which combines three potent agents. However, concerns about gastrointestinal side effects during the early phases of treatment have emerged, potentially impacting patient adherence and outcomes. Materials and Methods: This retrospective cohort study, conducted over four years in Romania, examined 222 patients initiated on BIKTARVY^®^ therapy. Data were collected from electronic medical records, and stringent inclusion and exclusion criteria were applied to ensure data accuracy and relevance. Statistical analysis was performed to assess age-related patterns in gastrointestinal symptoms and their relation with significant weight loss. Results: This study revealed significant differences in the prevalence of gastrointestinal symptoms between age groups, with older patients experiencing more symptoms. Notably, diarrhea did not exhibit a statistically significant age-related difference. Furthermore, weight loss exceeding 5 kg was more common in older patients. Of the patients who continued BIKTARVY^®^ therapy, 84.9% showed an increase in CD4 cell counts, and most expressed satisfaction with treatment. Conclusion: Understanding age-related patterns and gastrointestinal side effects of BIKTARVY^®^ is crucial for optimizing HIV patient care. Future research should aim to corroborate and expand upon these findings, potentially leading to improved therapeutic approaches in the ongoing fight against HIV.

## 1. Introduction

In the efforts to combat the human immunodeficiency virus (HIV), several therapeutic milestones have been discovered and have continuously evolved over the years [1]. Antiretroviral therapy (ART) remains the foundation in managing and curtailing the progression of HIV infection, leading to increased longevity and improved quality of life for individuals living with this chronic disease [2]. A typical initial regimen for HIV includes three HIV medications from at least two drug classes. Although this treatment is not curative, it can extend patients’ lives and reduce HIV transmission [3]. The combination of bictegravir, emtricitabine, and tenofovir alafenamide is a recent addition to the ART arsenal [3]. Since its introduction, it has gained attention for its promising efficacy and a relatively narrow side effect profile, especially when compared to previously used medications [4]. However, the field of medicine is a continuous journey of learning, and even the most lauded therapeutic agents are not immune to disadvantages. One such concern associated with “BIKTARVY^®^” (bictegravir, emtricitabine, and tenofovir alafenamide) is the gastrointestinal effects, especially in the early stages of treatment. Gastrointestinal disturbances, though they may seem minor, have profound implications [5]. These side effects can range from mild nausea to significant diarrhea or vomiting. The potential repercussions of these symptoms, if persistent, include malabsorption, dehydration, and electrolyte imbalances [6,7]. Moreover, the psychological and emotional impact on the patient cannot be underestimated, often leading to reduced adherence to medication, resulting in suboptimal treatment outcomes [8]. The relationship between the gastrointestinal side effects of this combination of drugs and the initial phase of therapy is intriguing. Gastrointestinal symptoms induced by medication are known to discourage patients from adhering to their regimen [9]. This is particularly concerning for antiretroviral agents, where consistent drug levels are imperative to prevent viral resistance. A disrupted therapeutic regimen not only endangers the health of the individual patient but also increases the risk of transmitting a potentially resistant strain of the virus to others [10]. Gastrointestinal symptoms that may be present at the beginning of therapy with these substances can lead to significant weight loss. A weight loss of more than 5 kg in a short period can be alarming [11]. Rapid weight loss may signal malnutrition, muscle wasting, or other underlying metabolic disturbances [11]. For HIV patients, who may already be dealing with issues of wasting, such a rapid decline in weight can have dire consequences.

Diving deeper, the age-related patterns of these adverse effects become a focal point of our investigation. Age is a complex variable in clinical research. Physiological changes, including altered drug metabolism, organ function, and the inherent resilience of the gastrointestinal tract, can vary significantly across different age categories [12]. It is plausible that certain age groups might bear a disproportionate burden of these side effects, or conversely, show increased resilience. In the management of HIV, understanding these nuances can have a profound impact on patient care. As we advance in this detailed analysis, our goal is to ensure the well-being of individuals living with HIV, striving to optimize their therapeutic experience and ensure that their journey with BIKTARVY^®^ is as smooth and beneficial as possible. The primary aim of this study is to assess the gastrointestinal adverse effects of BIKTARVY^®^ antiretroviral therapy during the first six months of treatment. This study specifically seeks to determine age-related patterns in these adverse effects and their relation with weight loss of more than 5 kg during the initial treatment period.

## 2. Materials and Methods

### 2.1. Study Design and Setting

This was a retrospective study conducted over a four-year period, examining patients initiated on BIKTARVY^®^ therapy across “Dr. Victor Babeș Clinical Hospital for Infectious Diseases and Pneumophthisiology” in Timisoara, Romania. This center was equipped with advanced diagnostic facilities, and patients were routinely monitored for any adverse effects post ART initiation.

### 2.2. Study Population

A total of 222 patients, both newly diagnosed and those switching from other ART regimens to BIKTARVY^®^, were included in this study. From the total number of patients included in this study, two groups were formed: group 1 consisting of patients aged between 18 and 45 years, and group 2 comprising patients over 45 years of age. The division of age categories in this manner was decided based on our experiences, where we have noticed changes in treatment response and the frequency of associated pathologies starting from the age of 46. At present, there are no relevant studies that show a documented division based on age category, but we intend to focus on this aspect. This research could signify a starting point in this direction.

The eligibility criteria included adult patients aged 18 years or older, exclusively following a treatment regimen with BIKTARVY^®^, without any other concurrent medications. Additionally, confirmation of a positive HIV diagnosis through standard serological testing was necessary, and patients could be either newly initiated on ART or transitioning from another ART regimen. Other essential criteria were the availability of comprehensive and reliable medical records, the absence of significant prior gastrointestinal pathologies, and not having received any other antiretroviral treatments in the last 30 days. The criteria also encompassed the ability to understand and provide informed consent for participation in the study and for women of childbearing age, a negative pregnancy test.

In medical practice, a weight loss or gain of around 5 kg is often considered a sensitive and significant indicator of health changes. This amount is large enough to be clinically relevant, yet not so large that it is rare among patient populations. Therefore, 5 kg serves as a useful benchmark for quickly identifying patients who may need further investigation or adjustments in their treatment. Setting a specific threshold of 5 kg for weight changes helps standardize patient monitoring and promptly identifies those who might benefit from nutritional interventions or treatment adjustments. This can be vital for maintaining overall health and long-term quality of life for patients. Additionally, antiretroviral therapy, including ‘Biktarvy’, can cause gastrointestinal side effects which in turn might affect appetite and nutrient absorption. A 5 kg weight change over a short period can be a sign of such side effects and therefore an important marker for assessing the drug’s tolerability.

The exclusion criteria were set to eliminate patients with untreated active co-infections such as tuberculosis or hepatitis B and C, known allergic reactions or intolerances to any of the three substances, severe renal insufficiency or abnormal renal function values, advanced hepatic insufficiency, or hepatic enzyme levels exceeding three times the upper normal limit. Also excluded were patients taking medications or supplements that interact with these substances and cannot be discontinued or substituted, those with severe cardiac conditions, a history of active malignancies or ones treated in the last 5 years (except for completely treated low-risk cancers), active drug or alcohol dependency not enrolled in a rehabilitation program, significant cognitive or neurological disorders, or the inability to comply with scheduled follow-up visits or other study requirements.

These criteria were established to ensure patient safety and to obtain the most accurate and relevant data for the study objectives. By excluding patients with potential complications or drug interactions, we were able to limit confounding variables and ensure that the study results directly reflect the impact of this therapy on patients. Furthermore, these exclusion criteria aid in protecting patients from potential risks or adverse effects that might arise from their participation in this study. Therefore, they contribute to the integrity and validity of this study, while also ensuring that patients are treated with the highest level of medical ethics and consideration.

### 2.3. Data Collection

To ensure that the study results were reliable and robust, a systematic approach to data collection was adopted. Our primary aim was to gather comprehensive patient data relating to BIKTARVY^®^ therapy, its effects, and any other relevant clinical and demographic details. Data were primarily extracted from the hospital’s centralized electronic medical record (EMR) system. This comprehensive system housed detailed patient profiles, including medical histories, medication regimens, laboratory test results, and documented adverse effects.

All collected data were anonymized to protect patient privacy. Identifiable details were replaced with unique identification codes. The data were then stored in a secure, encrypted database with multi-layered access controls, ensuring only authorized personnel could access and analyze the information. Regular backups were maintained to prevent data loss. Data extraction began in the month following the conclusion of the study’s four-year span and was completed within a three-month window to ensure timeliness and relevance in analysis.

Given the retrospective nature of this study, some challenges were anticipated, such as missing records or inconsistent documentation. To address these, a protocol was established to consult with the involved healthcare providers or to use secondary sources like pharmacy records to validate and complete the dataset.

In summary, meticulous planning and a systematic approach to data collection ensured that the information gathered was both comprehensive and of high quality, paving the way for meaningful analysis in subsequent stages of this study.

### 2.4. Statistical Analysis

In this study, we used GraphPad Prism 6 for our statistical evaluations. We employed two primary statistical tests to assess the significance of differences observed between our datasets: the *t*-test and the Z-test. The *t*-test is a parametric test designed to evaluate differences between the means of two groups. We opted for the *t*-test because our sample sizes were larger, and we aimed to compare results between two distinct samples. On the other hand, we chose the Z-test because our data were well defined and standardized. Additionally, we used the Z-test when we needed to compare two different proportions from the two groups (which are two separate samples), considering that for this type of data, the Z-test is more efficient than the T-test. The resulting *p*-value provides a quantitative measure of this, with a value typically less than 0.05 indicating statistically significant differences.

The Shapiro–Wilk test was used in data analysis to check the assumption of normality, which is crucial. This test examined the sample, which turned out to consist of data that follow a normal distribution and are suitable for the statistical analysis being used.

### 2.5. Ethical Considerations

This study was approved by the Institutional Review Board of the participating hospital (approval No. 8947/28 September 2018). Given the retrospective nature of this study, informed consent was waived. However, all patient data were anonymized and handled in compliance with the General Data Protection Regulation (GDPR) guidelines to ensure patient confidentiality.

## 3. Results

In the course of the four-year retrospective study, data were garnered from 222 patients who initiated BIKTARVY^®^ therapy at the designated hospital in Romania. The 222 patients were divided into two groups: group 1, consisting of patients aged between 18 and 45 years, and group 2, comprising patients aged over 45 years. The subsequent sections detail the primary findings of our investigation.

The demographic and socioeconomic characteristics of the two age groups are presented in Table 1. In the younger cohort, a higher proportion of males is evident, while the older group has a slight majority of females. It is noteworthy that in group 2, married patients predominate, whereas the younger group has a more varied marital status distribution, including a significant proportion of singles. Differences in education levels are also apparent, with a higher percentage of individuals in the older group possessing a Bachelor’s/College degree. Regarding occupation, retirees are exclusively found in the older group. Health disparities are striking: the younger group exhibits a significant percentage without any comorbidities, whereas the older age group shows higher incidences of hypertension, diabetes, cardiovascular diseases, and renal afflictions.

A significant majority of patients (192, 86.5%) continued with the studied therapy over a one-year period. In contrast, 30 patients (13.5%) chose to discontinue the therapy, mainly attributing this to side effects or personal considerations.

At the time of patient enrollment in the study, we measured their height and weight to calculate their BMI. Figure 1 shows the distribution of patients based on the weight category that they were classified into according to their BMI at the two key moments of the study (at the time of study enrollment and after 6 months of treatment). Table 2 presents a detailed analysis of the BMIs of patients from both groups at the two key moments of the study (at the time of study enrollment and after 6 months of treatment). Due to the *p*-value being over 0.05, we conclude that there is no statistically significant difference between the results of the two groups. It is noteworthy that after 6 months of treatment, even though there were fluctuations in weight, these were not significant enough for patients to be reclassified into a different weight category.

An analysis of the data reveals differences in symptom prevalence between the two age categories (Table 3). A higher incidence of appetite loss is observed in the older patient group, at 57.31% compared to 43.57%, with a statistical significance (*p* Value) of 0.017. Dysphagia, odynophagia, nausea, vomiting, and diffuse abdominal pain are also significantly more frequent in the over-45 age group, with statistically significant *p* values (all below 0.001), indicating a notable difference between the two age groups. On the other hand, while epigastric pain is more common in the older group, it does not show a statistically significant difference, unlike hematemesis, which reaches a *p* value of 0.044. Diarrhea also does not show a significant difference between groups, with a *p* value of 0.293.

All of the symptoms presented in the data are observed at a higher percentage in the older age group, and this difference is not merely coincidental. Statistically speaking, there is a significant disparity in the prevalence of these symptoms between the age groups, indicating that individuals of advanced age are more likely to experience these particular symptoms. This affirmation is supported by the *p*-values presented in the table.

Of the total 222 patients, 7 exhibited a weight loss exceeding 5 kg in the first month of treatment (Table 4). Notably, of these seven individuals, five belonged to the older age group, while two were from the younger patient category. These patients lost more than 5 kg of weight. The mean and SD in the table represent the average and standard deviation of the number of kilograms lost of weight, which is greater than 5 (group 1: 5.2 kg and 5.6 kg; group 2: 5.4 kg, 5.7 kg, 5.5 kg, 8.1 kg, and 6.1 kg). Upon comparing the weight loss percentages between these two age groups, a significant difference was discerned, with a *p*-value less than 0.001, indicating a statistically significant variation in weight loss between the two cohorts.

In Table 5, the weight losses and gains of less than 5 kg for both groups are described, along with the *p*-value, noting that there were no instances of weight gain exceeding 5 kg. This study revealed a statistically significant difference in weight maintenance between the two patient groups. The mean weight values and their standard deviations showed a close similarity, indicating a general trend of weight stability. Furthermore, a substantial portion of the patients in each group successfully maintained their weight within a defined range. The difference in the proportions of patients maintaining their weight between the two groups was notable and statistically significant, with a *p*-value of less than 0.001, suggesting a meaningful distinction in weight maintenance outcomes across the studied cohorts.

Within the initial month of therapy, 83 patients (37.38) reported experiencing gastrointestinal disturbances, ranking it as the foremost adverse effect. Subsequent side effects encompassed fatigue, noted in 35 patients (15.8%), and skin rashes, which affected 20 patients (9.0%). A total of 16 patients (7.2%) opted to cease the medication within the first month owing to these side effects.

Among those patients who continued the ‘Biktarvy’ therapy beyond the initial month, a remarkable 178 out of 210 (84.8%) experienced a notable increase in their CD4 cell counts by the end of the first year. It is noteworthy that for those who did not exhibit CD4 count improvements, treatment protocols were modified. These particular patients, while adhering to the medication regimen, sporadically missed their scheduled blood tests during the 12-month period. We consider the treatment successful and disease progression favorable in cases where there were observed increases in CD4 cell counts. For patients with increased CD4 counts, the viral load after one year was consistently below 35 IU/mL or undetectable.

At the onset of ‘Biktarvy’ treatment, CD4+ T-cell counts among patients varied widely. The initial minimum count was a concerning 5 cells/mm^3^. The lower quartile stood at 85 cells/mm^3^, indicating that a significant portion of patients had counts beneath this level. The median count was 135 cells/mm^3^, demonstrating an even split in the patient population in terms of higher and lower counts. The upper quartile was 175 cells/mm^3^, with the highest count recorded at 205 cells/mm^3^. Post one year of ‘Biktarvy’ treatment, the improvements were striking. The lowest count surged to an encouraging 190 cells/mm^3^, the lower quartile increased to 215 cells/mm^3^, and the median count climbed impressively to 275 cells/mm^3^. The upper quartile rose to 295 cells/mm^3^, and the highest observed count was an impressive 310 cells/mm^3^. This upward trend was also mirrored in the average CD4+ T-cell counts, which initially were 125.4 cells/mm^3^, with a standard deviation (SD) of 46.3, and enhanced to 250.2 cells/mm^3^, with an SD of 37.8 after one year, underscoring the therapy’s effectiveness in boosting CD4+ T-cell levels among patients (as illustrated in Figure 2).

Of the 192 patients who continued the therapy, 173 (90.1%) expressed satisfaction or high satisfaction with their treatment regimen. The main reasons underscoring this satisfaction included perceived therapeutic efficacy and the rapid diminishment of side effects.

As outlined earlier, 30 patients (13.5%) terminated the therapy. Beyond the realm of side effects, primary reasons that fueled discontinuation encompassed personal choices and external factors unrelated to the medication’s efficacy or side effects. Following a discussion with patients who chose not to discontinue BIKTARVY^®^ therapy, 93% of them reported a significant reduction in gastrointestinal adverse effects from the first month to the sixth month of treatment, stating that these effects were easily manageable.

## 4. Discussion

Selecting the optimal initial treatment regimen for a specific patient involves considering numerous factors. According to current guidelines, the recommended options typically involve using either lamivudine or emtricitabine in combination with zidovudine or tenofovir, along with efavirenz [13]. Alternatively, there is the option of a preferred protease inhibitor regimen, which includes atazanavir, fosamprenavir, saquinavir, or indinavir combined with lopinavir–ritonavir [14]. Other regimens are available as alternatives which encompass various combinations such as lamivudine or emtricitabine with zidovudine, stavudine, abacavir, tenofovir, or didanosine, all paired with a protease inhibitor (atazanavir, fosamprenavir, saquinavir, indinavir, or ritonavir), or in combination with a non-nucleoside reverse transcriptase inhibitor (NNRTI) like efavirenz, rilpivirine, or nevirapine [15].

When determining the choice of nucleoside reverse transcriptase inhibitor (NRTI) backbone, most patients initially receive one of the preferred combinations, unless the patient has either anemia or renal disease [16]. For patients with anemia, an alternative agent to zidovudine may be necessary [17], while those with renal disease may require an alternative agent to tenofovir or an adjustment in dosage [18]. In deciding between protease inhibitors and NNRTIs, the decision should take into account the adverse event profiles of NNRTIs [19].

Our study was an exhaustive exploration into the gastrointestinal adverse effects in patients undergoing BIKTARVY^®^ antiretroviral therapy during the initial six months of treatment. This medication is a widely used antiretroviral medication in the treatment of HIV-1 infection. Bictegravir is an integrase strand transfer inhibitor that effectively blocks the replication of the HIV virus within host cells. Emtricitabine and tenofovir alafenamide belong to the nucleoside reverse transcriptase inhibitor (NRTI) class, working in tandem to further inhibit HIV replication by interfering with the reverse transcription process. Together, bictegravir, emtricitabine, and tenofovir alafenamide make a potent and well tolerated option for managing HIV-1 infection, simplifying treatment regimens and improving patient adherence. Its effectiveness and relatively low risk of drug interactions have made it a valuable addition to the armamentarium of HIV therapies [20].

Previous studies have reported gastrointestinal side effects following therapy with this medication [21]. With a sample size of 222, this study offers considerable insights into how these effects manifest, especially in relation to age and significant weight loss.

One of the most pertinent observations was the variation in the occurrence of certain gastrointestinal symptoms between different age brackets. This research has demonstrated that gastrointestinal symptoms are more frequently observed in older adults compared to younger individuals. This suggests that age may play a role in susceptibility to these adverse effects, highlighting the need for heightened awareness and monitoring in the older patient population [22]. A retrospective analysis with the use of BIKTARVY^®^ was performed to analyze the factors associated with ≥10% body mass index increases versus <10% increase. The data showed that there were no significant changes in body mass index between pre and post switch to BIKTARVY^®^ time periods [23]. Notably, the older age group (those over 45) showed a higher prevalence of symptoms like loss of appetite, dysphagia, odynophagia, epigastric pain, and nausea compared to the younger age bracket. This might suggest that the older population could be more vulnerable to certain gastrointestinal side effects of BIKTARVY^®^. This difference in symptom occurrence, especially in gastrointestinal-related manifestations, can be attributed to various factors. Physiological changes associated with aging, such as reduced gastric motility, diminished gastric acid production, and alterations in the gut microbiome, might render the older population more susceptible to these side effects [24]. Moreover, other comorbidities commonly seen with aging might exacerbate these symptoms.

Another study found more weight gain among previously antiretroviral therapy-naïve people living with HIV on newer integrase strand transfer inhibitors (INSTIs) and ritonavir-boosted darunavir-based regimens in the current antiretroviral therapy treatment era, although weight gain appeared to plateau over time [25]. Our study demonstrated a statistical difference in weight maintenance between the two patient groups.

People living with HIV switching to BIKTARVY^®^ therapy experience weight gain after the first year of switching treatment. Although this weight gain could be due to the switch in treatment regimen, it cannot be excluded that it was caused by other factors, since no comparable control group could be used for comparison [26].

In the vast body of literature, diarrhea is frequently cited as a prevalent side effect linked to many antiretroviral therapies [27]. However, in the context of our findings, it is noteworthy that there was not a marked difference in the occurrence of diarrhea between the two age groups under consideration. This observation could suggest two possible interpretations. Firstly, it might imply that there is a fairly consistent vulnerability to this side effect across different age brackets. Alternatively, it could be a clue pointing toward the notion that this specific side effect might be independent of age and remains consistent regardless of the patient’s age.

A significant aspect that emerges from the data is the potential relationship between gastrointestinal disturbances and rapid weight loss. A decline of more than 5 kg in a short period is concerning, especially for HIV patients. This study’s findings underscore the importance of continuous monitoring and proactive management of these symptoms to prevent such drastic weight reductions. Early intervention, timely dietary modifications, or possible dose adjustments could be potential strategies to mitigate these effects.

The demographic and socioeconomic analysis offers another dimension of understanding. Factors like education, occupation, and area of residence can play a role in how patients perceive, manage, or even report these side effects [28]. For instance, the younger group’s higher educational attainment might result in better health literacy, influencing their adherence and response to therapy.

The findings from this study lead to multiple noteworthy clinical implications that deserve careful consideration. At the forefront is the necessity for healthcare professionals to approach the prescription of BIKTARVY^®^ for older patients with an elevated sense of caution and discernment. The unique vulnerabilities and physiological changes that tend to manifest with aging can make this demographic more susceptible to certain side effects.

Therefore, a specialized and individualized counseling approach becomes indispensable. During these sessions, emphasis should be placed on enlightening patients about potential gastrointestinal side effects. This detailed information equips patients with the knowledge that they need, allowing them to be more attentive to any anomalies and to communicate these promptly to their healthcare providers.

Moreover, adopting a regimen of regular and consistent monitoring is of paramount importance, with special focus during the initial phases of the therapeutic journey [29]. This vigilant oversight facilitates early identification of any emerging adverse reactions or complications. By recognizing these issues at their nascent stages, clinicians are in a better position to institute timely interventions, be it through mitigation strategies or adjustments in the treatment plan. Such a proactive stance not only safeguards the patient’s health but also enhances the overall efficacy and success of the therapy.

In the medical literature, there have been reports of other adverse events associated with this antiretroviral medication. Notably, cases of severe rhabdomyolysis and acute asymptomatic pancreatitis have been documented, particularly when BIKTARVY^®^ was used concomitantly in the context of a hyperosmolar diabetic crisis [30]. These findings underscore the importance of closely monitoring patients and being aware of potential complications when administering this medication.

Particularly, considering the age-related differences in symptoms like gastrointestinal issues and weight loss, our research underscores the need for healthcare professionals to carefully consider their approach when prescribing BIKTARVY^®^, especially for older patients. The higher occurrence of certain gastrointestinal symptoms in older individuals highlights the need for tailored treatment plans. This involves educating patients about possible side effects, regularly monitoring their health, and adjusting treatments based on each patient’s unique needs and reactions.

Additionally, the lack of significant differences in diarrhea incidents across age groups suggests that we need to rethink how this side effect is related to age. This underscores the importance of consistently monitoring gastrointestinal symptoms in all patients, not just the elderly.

Another key observation is the link between gastrointestinal disturbances and rapid weight loss in HIV patients. This calls for early interventions, like dietary changes and medication adjustments, to prevent negative health impacts.

Importantly, after one year of treatment with BIKTARVY^®^, there is an improvement in CD4 cell immunity and a favorable virusologic response.

Finally, our findings on the socioeconomic factors influencing patient experiences with BIKTARVY^®^ treatment highlight the necessity for more inclusive healthcare strategies. These should consider differences in health literacy and access to healthcare resources. By broadening our discussion to include these factors, we hope to contribute to a more comprehensive understanding of antiretroviral therapy’s effects and promote better patient care in various healthcare environments.

### Strengths and Limitation

Central to the credibility and relevance of our research are several strengths that set it apart from other studies in the field of antiretroviral therapy evaluations. Firstly, the extended duration of four years adds temporal depth to our study, ensuring the inclusion of a broad range of patients and enhancing the study’s generalizability. Additionally, the use of a centralized electronic medical record (EMR) system as the primary data source is another significant strength. This digital system provides us with a precise and structured dataset, facilitating accurate analyses. Furthermore, the substantial sample size of 222 patients lends statistical robustness. The explicit inclusion and exclusion criteria refine the patient cohort to be more homogeneous, reducing confounding variables and ensuring that the observed effects can confidently be attributed to the three examined substances: bictegravir, emtricitabine, and tenofovir alafenamide. This study’s comprehensive examination of both age-related patterns and weight change dynamics offers a more complete clinical picture than studies focusing on a single aspect or outcome. Finally, this study’s acknowledgment of its own limitations demonstrates transparency and academic rigor, enhancing its trustworthiness. Overall, these strengths, rooted in methodological rigor, strategic focus, and comprehensive analyses, position this study as a seminal contribution to understanding the gastrointestinal effects of BIKTARVY^®^ in the early stages of therapy.

Despite providing essential insights into the gastrointestinal adverse effects of antiretroviral therapy and its association with significant weight loss, several limitations must be acknowledged. Firstly, being conducted at a single hospital in Romania means that this study’s findings might not be universally applicable. Hospitals vary in their patient populations, treatment protocols, and care quality, so the results might reflect the specific characteristics of the institution rather than broader patterns across multiple healthcare settings. Another concern is this study’s retrospective design. This approach inherently depends on past records, which may have inconsistencies, missing data points, or undocumented adverse effects, leading to potential biases or incomplete representations of the actual clinical scenario. While our team was diligent in addressing missing or ambiguous data, retrospective studies cannot offer the same level of control over variables as prospective ones.

Moreover, given this study’s focus on the first six months of therapy, long-term effects or late-onset complications related to BIKTARVY^®^ might not have been captured. This six-month window may not fully encompass the entirety of gastrointestinal responses, especially delayed reactions or cumulative effects that might manifest beyond the initial treatment period.

## 5. Conclusions

To summarize our findings, our study reveals that BIKTARVY^®^, though an effective antiretroviral therapy, may cause gastrointestinal side effects, especially in some age groups. It is essential for healthcare professionals to be aware of these aspects to customize patient care for the best possible outcomes. This knowledge is vital for our continued quest to improve life for those affected by HIV.

Looking ahead, we suggest conducting more research with a forward-looking, prospective approach, involving larger and more varied groups of people. Such research could shed more light on the specific effects of BIKTARVY^®^ and help fine-tune treatment plans for a wider range of patients. Investigating the long-term consequences of this therapy and how it interacts with other health conditions could also offer crucial insights for more holistic HIV treatment and care.

## Figures and Tables

**Figure 1 diseases-12-00003-f001:**
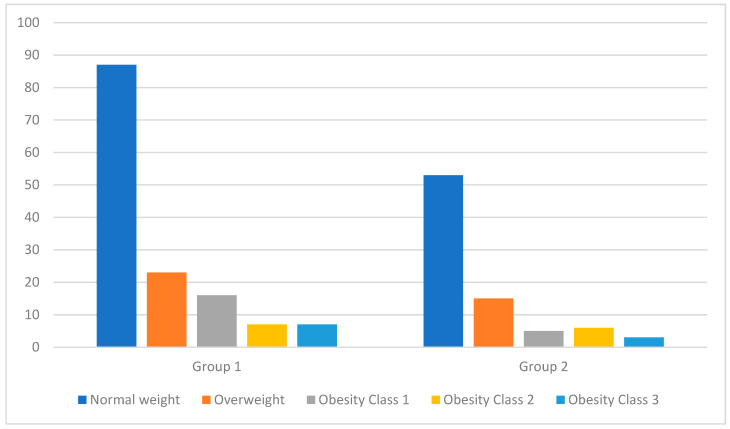
Distribution of patients by weight category based on BMI classification at the time of patient enrollment in the study.

**Figure 2 diseases-12-00003-f002:**
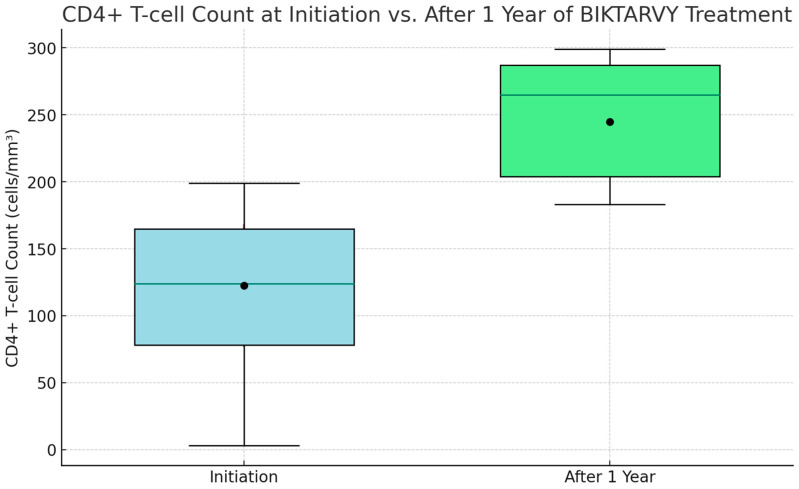
CD4+ T-cell count at initiation vs. after 1 year of BIKTARVY^®^ treatment.

**Table 1 diseases-12-00003-t001:** Demographic and socioeconomic characteristics by age group.

Demographic Criteria	Group 1 (Patients Aged between 18 and 45 Years)	Group 2 (Patients Aged over 45 Years)	*p* Value
Gender:			
- Male	80 (57.1%)	40 (48.8%)	0.01
- Female	60 (42.9%)	42 (51.2%)	0.009
Marital Status:			
- Married	65 (46.4%)	75 (91.5%)	<0.001
- Single	60 (42.9%)	0 (0%)	<0.001
- Divorced	10 (7.1%)	5 (6.1%)	0.277
- Widowed	5 (3.6%)	2 (2.4%)	0.0322
Education:			
- No formal education	5 (3.6%)	5 (6.1%)	0.0001
- High School	70 (50%)	30 (36.6%)	<0.001
- Bachelor’s/College Degree	50 (35.7%)	40 (48.8%)	<0.001
- Post-Graduate Studies	15 (10.7%)	7 (8.5%)	<0.001
Occupation:			
- Unemployed	10 (7.1%)	5 (6.1%)	0.0002
- Skilled Worker	50 (35.7%)	20 (24.4%)	<0.001
- Professional	50 (35.7%)	10 (12.2%)	<0.001
- Retiree	5 (3.6%)	72 (87.8%)	<0.001
Comorbidities:			
- Hypertension	15 (10.7%)	40 (48.8%)	<0.001
- Diabetes	10 (7.1%)	20 (24.4%)	<0.001
- Cardiovascular Diseases	10 (7.1%)	40 (48.8%)	<0.001
- Renal Afflictions	5 (3.6%)	10 (12.2%)	<0.001
- Without Comorbidities	100 (71.4%)	0 (0%)	<0.001

**Table 2 diseases-12-00003-t002:** Comparison of body mass index (BMI) across age groups of patients at the time of study enrollment.

BMI	At the Time of Study Enrollment		After 6 Months of Treatment	
	Group 1 (Patients Aged between 18 and 45 Years)	Group 2 (Patients Aged over 45 Years)	Group 1 (Patients Aged between 18 and 45 Years)	Group 2 (Patients Aged over 45 Years)
Minimum	19.05	19.33	19.05	19.33
25% Percentile	21.85	21.35	21.93	21.32
Median	24.10	23.83	24.11	23.66
75% Percentile	26.89	26.92	27.02	26.92
Maximum	43.82	44.82	43.99	44.62
Mean ± SD	25.78 ± 5.941	25.53 ± 5.816	25.94 ± 6.114	25.49 ± 5.937
*p* Value	0.4453		0.5899	

**Table 3 diseases-12-00003-t003:** Comparative prevalence of gastrointestinal symptoms across two age demographics.

	Group 1 (Patients Aged between 18 and 45 Years)	Group 2 (Patients Aged over 45 Years)	*p* Value
Loss of appetite	61 (43.57%)	47 (57.31%)	0.017
Dysphagia	11 (7.85%)	32 (39.02%)	<0.001
Odynophagia	13 (9.28%)	41 (50%)	<0.001
Epigastric pain	9 (6.42%)	11 (13.41%)	0.057
Nausea	14 (10%)	29 (35.36%)	<0.001
Vomiting	9 (6.42%)	17 (20.73%)	<0.001
Diffuse abdominal pain	26 (18.57%)	31 (37.80%)	<0.001
Hematemesis	2 (1.42%)	5 (6.09%)	0.044
Diarrhea	18 (12.85%)	14 (17.07%)	0.293

**Table 4 diseases-12-00003-t004:** Comparison of weight loss exceeding 5 kg in group 1 vs. group 2.

Weight Loss > 5 kg	Group 1 (Patients Aged between 18 and 45 Years)	Group 2 (Patients Aged over 45 Years)
Mean ± SD	5.400 ± 0.2828 kg	6.160 ± 1.117 kg
Number of patients	2 (1.42%)	5 (6.09%)
*p*-value (for weight loss)	<0.001	

**Table 5 diseases-12-00003-t005:** Comparison of weight changes (maintenance, loss, and gain) less than 5 kg and related *p*-values in patient cohorts.

Criteria	Group 1 (Patients Aged between 18 and 45 Years)	Group 2 (Patients Aged over 45 Years)
Weight loss ≤ 5 kg		
Mean ± SD	2.050 ± 1.229	1.661 ± 1.253
Number of patients	32 (22.85%)	28 (34.14%)
*p*-value	<0.001	
Weight gained ≤ 5 kg		
Mean ± SD	2.238 ± 0.7877	2.217 ± 0.8229
Number of patients	32 (22.85%)	29 (35.36%)
*p*-value	<0.001	
Maintained weight		
Mean ± SD	77.01 ± 16.79	76.78 ± 16.64
Number of patients	74 (52.85%)	20 (24.39%)
*p*-value	<0.001	

## Data Availability

The datasets used and/or analyzed during the current study are available from the first author.

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
