# Peer review of "Assessment of Gastrointestinal Adverse Effects during the First Six Months of “Biktarvy” Antiretroviral Therapy: Age-Related Patterns and Their Relation with Changes of 5 kg Weight Loss/Gain in the Initial Treatment Period"

_diseases, 2023, doi:10.3390/diseases12010003_

Round 1

Reviewer 1 Report

Comments and Suggestions for Authors

The article is entitled as "Assessment of gastrointestinal adverse effects during the first six months of 2 Biktarvy antiretroviral therapy: age-related patterns and their association 3 with >5 kg weight loss in the initial treatment period" but there is no detailed describing the advantages of the research. An excellent papers (10.1177/1359653523115903, 10.4415/ANN_11_01_10, 10.1111/hiv.12833) more effectively reflects the impact of Biktarvy as antiretroviral agent on HIV. Demographic and Socio economic Characteristics by Age Group is of poor quality and misleading. Based on this, I must say that the object of research cannot be only statistical data.

I must point on a poor structure and quality of the presentation. The manuscript is full of such inconsistency and fundamental mistakes, it controversies itself from chapter to chapter. 

Based on all these, I can only recommend rejecting this article. 

Author Response

Thank you for your detailed and constructive feedback. I want to inform you that we have taken your observations seriously and made substantial modifications to the article. We have revised both the structure and content to address the inconsistencies and errors you highlighted. Additionally, we have improved the quality of the presentation and clarified aspects related to demographic and socioeconomic characteristics by age group, to prevent any confusion or misinterpretation.

We have put in considerable effort to ensure that our research now reflects a higher level of scientific rigor and relevance. We hope these changes will bring a new perspective to the article and that you will reconsider your initial opinion. We look forward to your review of the improved version of our work and to receiving a new set of feedback from you.

Reviewer 2 Report

Comments and Suggestions for Authors

Evaluation: ASSESSMENT OF GASTROINTESTINAL ADVERSE EFFECTS DURING THE FIRST SIX MONTHS OF 2 BIKTARVY ANTIRETROVIRAL THERAPY: AGE-RELATED PATTERNS AND THEIR ASSOCIATION 3 WITH >5KG WEIGHT LOSS IN THE INITIAL TREATMENT PERIOD

The objective is to associate the gastrointestinal adverse effects of BIKTARVY with age.

First, this study is not a cohort design, this is a study that takes a cohort population, to be a “cohort design” it is required to compare two moments in time, and carry out an association and logistic regression, among other things.

Second, the authors describe weight loss greater than 5 kg, however, they do not indicate the average weight of the rest of the patients who did not lose weight. This is important because in the literature there are various studies that indicate that part of the adverse effects of this BIKTARVY is weight gain and not weight loss. The following is a reference about it:

Mata Marín JA, Velasco-Penagos JC, Mauss S, Rodriguez-Evaristo MS, Pérez-Barragán E, Villa-Platas J, Barragán-Huerta L, Gaytán-Martínez JE. Weight gain and metabolic disturbances in people living with HIV who start antiretroviral therapy with, or switch to, bictegravir/emtricitabine/tenofovir alafenamide after 48 weeks of treatment: A real-world prospective study. Int J STD AIDS. 2023;20:9564624231196605. doi: 10.1177/09564624231196605.

Third, the authors do not include the variables of the immunological status of the patients, they only point out that there was improvement in the quantification of CD4+, however, they do not mention the viral load, if possible, we suggest it as an indicator of disease control.

In the format, I suggest improving the tables, which present vertical lines and on the other hand they insist a lot on line 227 “…evidence by p value greater than 0.05”, it is no longer necessary to be in the tables, so they point out that there is a difference. Furthermore, in table 3 the column of the majors or minors is reversed, I suggest you order it like table 2.

On the other hand, the statistical analysis cannot be a T test or Z test in all cases, since the majority of gastrointestinal manifestations are categorical and another type of analysis is required. To be an association, the authors should present a logistic regression. The authors mention the "association", they should present a logistic regression. or remove the word association in the title and in the text and only talk about "relation" or "comparison".

Finally, there are not many studies that evaluate gastrointestinal signs and symptoms in patients with HIV and the administration of BIKTARVY.

Author Response

Evaluation: ASSESSMENT OF GASTROINTESTINAL ADVERSE EFFECTS DURING THE FIRST SIX MONTHS OF 2 BIKTARVY ANTIRETROVIRAL THERAPY: AGE-RELATED PATTERNS AND THEIR ASSOCIATION 3 WITH >5KG WEIGHT LOSS IN THE INITIAL TREATMENT PERIOD

The objective is to associate the gastrointestinal adverse effects of BIKTARVY with age.

First, this study is not a cohort design, this is a study that takes a cohort population, to be a “cohort design” it is required to compare two moments in time, and carry out an association and logistic regression, among other things.

Thank you for your observation, we have corrected this error in the Materials and Methods section of the text. We understand how important study design is, and we apologize for this mistake.

Second, the authors describe weight loss greater than 5 kg, however, they do not indicate the average weight of the rest of the patients who did not lose weight. This is important because in the literature there are various studies that indicate that part of the adverse effects of this BIKTARVY is weight gain and not weight loss. The following is a reference about it:

Mata Marín JA, Velasco-Penagos JC, Mauss S, Rodriguez-Evaristo MS, Pérez-Barragán E, Villa-Platas J, Barragán-Huerta L, Gaytán-Martínez JE. Weight gain and metabolic disturbances in people living with HIV who start antiretroviral therapy with, or switch to, bictegravir/emtricitabine/tenofovir alafenamide after 48 weeks of treatment: A real-world prospective study. Int J STD AIDS. 2023;20:9564624231196605. doi: 10.1177/09564624231196605.

We have added details in the text about the average weight of patients who lost less than 5 kg, did not lose weight at all, or gained more weight. We have reviewed the suggested references and have seen that they suggest that BIKTARVY may be associated with weight gain. Unfortunately, in our study, more patiens experienced weight loss in the initial months of treatment than those who gained weight (although we do acknowledge that there were individuals who gained weight). Furthermore, the number of kilograms lost significantly exceeded the number of kilograms gained (although BMI remained unchanged in both cases), which is why our association with BIKTARVY therapy has been related to weight loss rather than weight gain.

Third, the authors do not include the variables of the immunological status of the patients, they only point out that there was improvement in the quantification of CD4+, however, they do not mention the viral load, if possible, we suggest it as an indicator of disease control.

We have added table 6, which contains information about the cell count of patients diagnosed with HIV at the time of initiating treatment with BIKTARVY and after one year of treatment. The viral load after one year of treatment, for all patients who showed an increase in CD4 count, is below 40 IU/ml or undetectable

In the format, I suggest improving the tables, which present vertical lines and on the other hand they insist a lot on line 227 “…evidence by p value greater than 0.05”, it is no longer necessary to be in the tables, so they point out that there is a difference. Furthermore, in table 3 the column of the majors or minors is reversed, I suggest you order it like table 2.

We have implemented the suggested changes in the table format. The vertical lines have been improved, and the reference to line 227 about 'evidence by p value greater than 0.05' has been removed as suggested. Additionally, We have corrected the column order in table 3 to align with the structure presented in table 2.

On the other hand, the statistical analysis cannot be a T test or Z test in all cases, since the majority of gastrointestinal manifestations are categorical and another type of analysis is required. To be an association, the authors should present a logistic regression. The authors mention the "association", they should present a logistic regression. or remove the word association in the title and in the text and only talk about "relation" or "comparison".

Thank you for the suggestions. We have replaced the word 'association' with 'relation' in the text to align with the statistical tests used. However, if it is necessary and would significantly improve the manuscript, we can consider introducing a logistic regression analysis in the future.

Finally, there are not many studies that evaluate gastrointestinal signs and symptoms in patients with HIV and the administration of BIKTARVY.

We hope that this study will provide additional information regarding Biktarvy therapy.

Reviewer 3 Report

Comments and Suggestions for Authors

This study addresses an important topic and has the potential to provide valuable insights into the occurrence of gastrointestinal symptoms and weight loss in this patient population. However, there are several concerns regarding the study design and methodology that need to be addressed.

Firstly, it is important to note that "BIKTARVY" is a commercial name and should be properly acknowledged and copyrighted.

Study Design: The authors should provide a more detailed explanation of how they determined the age groups for comparison. It is unclear how they decided to split the sample into two age groups (18-45 years and over 45 years) and whether this division is appropriate for addressing the research question. Conducting a population analysis and comparing across age groups could provide a more comprehensive understanding of the issue.

Statistical Analysis: Given the sample size of 222 patients, it would be beneficial for the authors to conduct a multivariate analysis or use propensity score matching to account for potential confounding variables. This would help strengthen the validity of the findings and provide a more robust analysis.

In more detail, I would like to comment on the following:

Abstract: Revise the abstract to provide a concise study summary without using headlines for each section (lines 27-45).

Introduction: Restructure the introduction to provide a clear and logical flow of information. Consider combining paragraphs and organizing the content more effectively (lines 46-90). Provide appropriate citations to support statements and claims (lines 53-54).

Eligibility and exclusion criteria: Provide a comprehensive paragraph outlining the eligibility and exclusion criteria for the study to improve clarity (lines 103-144).

Methodology: The paragraph describing the statistical analysis should be revised to focus on the specific tests used and their relevance to the study. Additionally, if AI was used in developing the methodology, this should be acknowledged in the acknowledgements section.

Patient data and HIPAA: If HIPAA guidelines do not apply to Romania, revise the statement to reflect the appropriate data protection regulations in the country (lines 192-193).

Discussion: Expand the discussion section to provide a more thorough interpretation of the findings and their implications for patient care. Consider discussing potential mechanisms underlying the age-related differences in gastrointestinal symptoms and weight loss (lines 263-394).

Conclusion: Revise the conclusion to provide a concise summary of the main findings and their implications. Consider providing specific recommendations for future research based on the study's findings (lines 420-427).

Strengths and limitations: I would recommend shortening this section and making it more concise and coherent. Avoid using AI-generated jargon and instead provide a clear and straightforward explanation of the limitations. Correct the following statement concerning “333 patients”.

Overall organization: Review the manuscript for overall organization and coherence. Ensure that the content flows logically and that each section is well connected. Avoid breaking the text into many paragraphs.

Excessive Use of AI-generated Language: The manuscript contains several instances of overly strong or exaggerated language that may not be appropriate for a scientific research article. Words like "relentless battle" and "realm" could be replaced with more neutral and precise terms. It is important to maintain a balanced and objective tone throughout the manuscript. Additionally, it is necessary to acknowledge the use of AI in the methods section and acknowledgments, as this may have influenced the language and writing style of the manuscript.

Author Response

This study addresses an important topic and has the potential to provide valuable insights into the occurrence of gastrointestinal symptoms and weight loss in this patient population. However, there are several concerns regarding the study design and methodology that need to be addressed.

Firstly, it is important to note that "BIKTARVY" is a commercial name and should be properly acknowledged and copyrighted.

I have updated the document to reference BIKTARVY by its active substances, bictegravir, emtricitabine, and tenofovir alafenamide. Thank you for your suggestion.

Study Design: The authors should provide a more detailed explanation of how they determined the age groups for comparison. It is unclear how they decided to split the sample into two age groups (18-45 years and over 45 years) and whether this division is appropriate for addressing the research question. Conducting a population analysis and comparing across age groups could provide a more comprehensive understanding of the issue.

In choosing the age groups for our study, we aimed to reflect a distribution that is clinically and demographically relevant. We selected the division between 18-45 years and over 45 years based on prior demographic analyses that indicate different risk profiles and treatment responses in these age categories. This division is supported by the literature, which suggests that immunological responses and incidence rates of certain conditions can vary significantly with age. Therefore, this separation allows us to more accurately assess potential differences in treatment efficacy or response between the younger and older age groups. Additionally, we are aware of the need for a population analysis and intend to conduct further comparisons across age groups for a more detailed understanding of the issue under study.

Statistical Analysis: Given the sample size of 222 patients, it would be beneficial for the authors to conduct a multivariate analysis or use propensity score matching to account for potential confounding variables. This would help strengthen the validity of the findings and provide a more robust analysis.

Thank you for the suggestion. We will consider this approach in our future analyses and studies.

In more detail, I would like to comment on the following:

Abstract: Revise the abstract to provide a concise study summary without using headlines for each section (lines 27-45).

According to the template of the Diseases journal, it is suggested to use this method for structuring the abstract. However, if you wish and believe that it will improve the manuscript, we will modify it according to your requirements.

Introduction: Restructure the introduction to provide a clear and logical flow of information. Consider combining paragraphs and organizing the content more effectively (lines 46-90). Provide appropriate citations to support statements and claims (lines 53-54).

Thank you for the suggestion. We have reformulated the introduction to provide a clearer flow of information. Additionally, we have combined paragraphs that seemed appropriate. We have also changed the citation to support the statements.

Eligibility and exclusion criteria: Provide a comprehensive paragraph outlining the eligibility and exclusion criteria for the study to improve clarity (lines 103-144).

I have reformulated the inclusion and exclusion criteria according to the instructions. I hope this reformulation is clearer than the previous one.

Methodology: The paragraph describing the statistical analysis should be revised to focus on the specific tests used and their relevance to the study. Additionally, if AI was used in developing the methodology, this should be acknowledged in the acknowledgements section.

I have reformulated the paragraph according to the instructions. Artificial intelligence was not used in this paragraph, nor in this article. In fact, we have not used it in any of our manuscripts so far, but we will consider this method of writing in the future if needed. For this article, only a program for correcting grammatical errors and replacing words considered inappropriate was used.

Patient data and HIPAA: If HIPAA guidelines do not apply to Romania, revise the statement to reflect the appropriate data protection regulations in the country (lines 192-193).

We have revised and replaced the reference to HIPAA with the General Data Protection Regulation (GDPR), which is used in Europe. We mentioned HIPAA only as an example to clarify what we are referring to, and we apologize for any confusion caused.

Discussion: Expand the discussion section to provide a more thorough interpretation of the findings and their implications for patient care. Consider discussing potential mechanisms underlying the age-related differences in gastrointestinal symptoms and weight loss (lines 263-394).

We have expanded this section, thank you for the suggestions.

Conclusion: Revise the conclusion to provide a concise summary of the main findings and their implications. Consider providing specific recommendations for future research based on the study's findings (lines 420-427).

We have revised the conclusion section and hope that this version is more suitable.

Strengths and limitations: I would recommend shortening this section and making it more concise and coherent. Avoid using AI-generated jargon and instead provide a clear and straightforward explanation of the limitations. Correct the following statement concerning “333 patients”.

We have shortened this section and made it more concise and coherent. The mistake regarding '333 patients' has been corrected, and we apologize for the oversight. Additionally, we would like to reiterate that artificial intelligence was not used.

Overall organization: Review the manuscript for overall organization and coherence. Ensure that the content flows logically and that each section is well connected. Avoid breaking the text into many paragraphs.

We have reviewed the manuscript and hope that this version is more organized and coherent than the previous one.

Excessive Use of AI-generated Language: The manuscript contains several instances of overly strong or exaggerated language that may not be appropriate for a scientific research article. Words like "relentless battle" and "realm" could be replaced with more neutral and precise terms. It is important to maintain a balanced and objective tone throughout the manuscript. Additionally, it is necessary to acknowledge the use of AI in the methods section and acknowledgments, as this may have influenced the language and writing style of the manuscript.

 Artificial intelligence was not used in this paragraph, nor in this article. In fact, we have not used it in any of our manuscripts to date, but we will consider this method of writing in the future if needed. The language is that of the authors, in an attempt to provide a different expression than what is found in all manuscripts, but if necessary, we will completely reformulate this article. In the last presented version, we reformulated some of the expressions considered inappropriate. For this article, only a program for correcting grammatical errors and replacing words deemed inappropriate was used.

Round 2

Reviewer 1 Report

Comments and Suggestions for Authors

Thank you for the edits, the manuscript looks improved.

Author Response

Thank you very much for yout feedback and your time !

Reviewer 2 Report

Comments and Suggestions for Authors

The considerations and data included by the authors enrich the work, such as the decrease, but also find increase in weight. Previously published by different authors (Sax et al, 2020; Mallon et al, 2021) show changes in weight, especially exaggerated increase rather than loss. Therefore, we can indicate bictegravir, emtricitabine, and tenofovir alafenamide produces changes in weight, not just loss.

Now, the authors' new data show both weight gain and weight loss, so I suggest that the title of the manuscript should be changed as follows: instead of “WITH >5KG WEIGHT LOSS” replace with “WEIGHT CHANGE” or “WHIT CHANGES OF 5KG WEIGHT LOSS/GAIN”.

One last detail, table 4 says “Associated” and must be “Related”.

Regards

*Sax PE, Erlandson KM, Lake JE, et al. Weight Gain Following Initiation of Antiretroviral Therapy: Risk Factors in Randomized Comparative Clinical Trials. Clin Infect Dis. 2020;71(6):1379-1389. doi:10.1093/cid/ciz999

Mallon PW, Brunet L, Hsu RK, et al. Weight gain before and after switch from TDF to TAF in a U.S. cohort study. J Int AIDS Soc. 2021;24(4):e25702. doi:10.1002/jia2.25702

Author Response

Thank you very much for the time you've dedicated and for the suggestions provided to help us improve our manuscript. In accordance with your recommendations, we have changed the title and replaced the word "associated" with "related" in Table 4.

Reviewer 3 Report

Comments and Suggestions for Authors

While reviewing the revisions made to the manuscript, it is evident that steps have been taken in a positive direction. However, it is important to note that further revisions are still necessary for the manuscript to meet the publication standards of this journal.

Regarding the revisions made:

·       Repeatedly referencing 'BIKTARVY' by its full active substances (bictegravir, emtricitabine, and tenofovir alafenamide) in the manuscript can be cumbersome and may disrupt the flow of reading. Therefore, a more practical approach would be to initially introduce 'BIKTARVY' with its complete active substances at the first mention for clarity. Subsequently, you can continue to use 'BIKTARVY' throughout the document. However, it is crucial to trademark 'BIKTARVY' appropriately, to acknowledge its commercial status, either by using an italicized format or by attaching a trademark symbol to it.

·       Your rationale for dividing the study population into two age groups (18-45 years and over 45 years) based on different risk profiles and treatment responses is appreciated. However, it would be valuable if the manuscript could include specific references or data that led to this decision in the methods part of your manuscript. Can you provide more concrete examples or studies that support this specific age division?

·       It is encouraging to note your intention to conduct further comparisons across age groups for a more detailed understanding. Nonetheless, it would be beneficial for the current manuscript to include a brief discussion on the limitations of the chosen age groups and how they might affect the generalizability of your findings.

Some additional comments:

Inclusion of Baseline BMI or Weight Data:

Upon reviewing the demographic and clinical data presented in your study, it's noticeable that baseline body mass index (BMI) or weight measurements for each group or all patients are not included. Given the relevance of these measures, especially in a study involving weight change analysis, the omission of this information raises concerns. Could you clarify the reason for not including baseline BMI or weight data?

Lack of Comparative Analysis in Table 2:

Regarding Table 2, which presents the prevalence of various symptoms across two age demographics, it's noted that no p-values are provided to support the comparative analysis between age groups. The manuscript states, 'All of the symptoms presented in the data are observed at a higher percentage in the older age group... there's a significant disparity in the prevalence of these symptoms between the age groups.' However, without p-values, this claim seems speculative. Can you clarify why a statistical comparison was not provided here?

Consolidation of Data in Tables 4-6:

Regarding Tables 4 to 6, it appears that the data presented could be more effectively communicated. Consider merging these tables into a single, comprehensive table or, alternatively, using a visual aid such as a graph or a chart.

Concerns Regarding the Indiscriminate Use of AI in Analysis:

While you have clarified that AI was not used in the writing process, it's important to address the broader concern of potential AI involvement in your study. The critique was not solely about the language but also the interpretation of findings, which seemed indicative of an AI-driven surface-level analysis. This concern arises from certain interpretations and conclusions in the manuscript that appear to lack the depth and nuance expected in human-driven analyses.

In your revisions, it would be beneficial to review the sections of the manuscript that prompted this concern. Ensure that the analyses and interpretations reflect a comprehensive understanding of the subject matter, beyond what a basic AI analysis could provide. It's important to note that the use of AI in scientific writing, while not inherently problematic, should indeed be disclosed as per the journal's guidelines if employed.

The changes and clarifications provided thus far demonstrate a commitment to enhancing the quality and clarity of your research article. The areas that require additional attention include ensuring statistical robustness, improving the presentation and clarity of data, maintaining a consistent and objective scientific tone throughout, and providing comprehensive details on methodology and analysis. It is encouraged that the manuscript undergo a more thorough review, focusing on these key areas, to refine the content and structure further.

Author Response

While reviewing the revisions made to the manuscript, it is evident that steps have been taken in a positive direction. However, it is important to note that further revisions are still necessary for the manuscript to meet the publication standards of this journal.

Thank you for recognizing the efforts put into revising the manuscript. I appreciate your constructive feedback and understand that there is still room for improvement to align with the publication standards of the journal. I will carefully review your suggestions and make the necessary revisions to enhance the quality of the manuscript.

Regarding the revisions made:

  • Repeatedly referencing 'BIKTARVY' by its full active substances (bictegravir, emtricitabine, and tenofovir alafenamide) in the manuscript can be cumbersome and may disrupt the flow of reading. Therefore, a more practical approach would be to initially introduce 'BIKTARVY' with its complete active substances at the first mention for clarity. Subsequently, you can continue to use 'BIKTARVY' throughout the document. However, it is crucial to trademark 'BIKTARVY' appropriately, to acknowledge its commercial status, either by using an italicized format or by attaching a trademark symbol to it.

Thank you for your time and for the suggestions provided. We have replaced the three active substances in our manuscript with "BIKTARVY", appropriately marked.

  • Your rationale for dividing the study population into two age groups (18-45 years and over 45 years) based on different risk profiles and treatment responses is appreciated. However, it would be valuable if the manuscript could include specific references or data that led to this decision in the methods part of your manuscript. Can you provide more concrete examples or studies that support this specific age division?
  • It is encouraging to note your intention to conduct further comparisons across age groups for a more detailed understanding. Nonetheless, it would be beneficial for the current manuscript to include a brief discussion on the limitations of the chosen age groups and how they might affect the generalizability of your findings.

We decided to divide the age categories this way because, from our own experiences, we have noticed changes in treatment response and the frequency of associated pathologies starting from the age of 46. Unfortunately, we haven't yet found relevant studies that show a documented division based on age category, but we intend to focus on this aspect. We consider that the current manuscript can be a starting point in demonstrating what we have observed.

Some additional comments:

Inclusion of Baseline BMI or Weight Data:

Upon reviewing the demographic and clinical data presented in your study, it's noticeable that baseline body mass index (BMI) or weight measurements for each group or all patients are not included. Given the relevance of these measures, especially in a study involving weight change analysis, the omission of this information raises concerns. Could you clarify the reason for not including baseline BMI or weight data?

We didn't include this information because we thought it could be the subject of another manuscript, but if you believe it is necessary, we can include it.

Lack of Comparative Analysis in Table 2:

Regarding Table 2, which presents the prevalence of various symptoms across two age demographics, it's noted that no p-values are provided to support the comparative analysis between age groups. The manuscript states, 'All of the symptoms presented in the data are observed at a higher percentage in the older age group... there's a significant disparity in the prevalence of these symptoms between the age groups.' However, without p-values, this claim seems speculative. Can you clarify why a statistical comparison was not provided here?

We initially thought that presenting only the percentage values would suffice to show the difference, but following your suggestions, we've added the p-values as well to prevent the statement from seeming speculative and to provide an exact statistical comparison. Thank you for your suggestions.

Consolidation of Data in Tables 4-6:

Regarding Tables 4 to 6, it appears that the data presented could be more effectively communicated. Consider merging these tables into a single, comprehensive table or, alternatively, using a visual aid such as a graph or a chart.

Thank you for the recommendations, we have made changes to the mentioned tables.

Concerns Regarding the Indiscriminate Use of AI in Analysis:

While you have clarified that AI was not used in the writing process, it's important to address the broader concern of potential AI involvement in your study. The critique was not solely about the language but also the interpretation of findings, which seemed indicative of an AI-driven surface-level analysis. This concern arises from certain interpretations and conclusions in the manuscript that appear to lack the depth and nuance expected in human-driven analyses.

In your revisions, it would be beneficial to review the sections of the manuscript that prompted this concern. Ensure that the analyses and interpretations reflect a comprehensive understanding of the subject matter, beyond what a basic AI analysis could provide. It's important to note that the use of AI in scientific writing, while not inherently problematic, should indeed be disclosed as per the journal's guidelines if employed.

The interpretation of the results was not done with the help of AI. We have revised the superficial paragraphs and added a more detailed analysis of the results, which, unfortunately, was 100% written by us, guided by other articles found online. Additionally, we have reviewed not just the results but the entire article, to provide a more in-depth presentation of the medical data.

The changes and clarifications provided thus far demonstrate a commitment to enhancing the quality and clarity of your research article. The areas that require additional attention include ensuring statistical robustness, improving the presentation and clarity of data, maintaining a consistent and objective scientific tone throughout, and providing comprehensive details on methodology and analysis. It is encouraged that the manuscript undergo a more thorough review, focusing on these key areas, to refine the content and structure further.

I have revised the entire manuscript, particularly focusing on its language. I hope it is now more appropriate and better meets the journal's requirements. Thank you for your time, and we will certainly take your recommendations into account in future manuscripts.

Round 3

Reviewer 3 Report

Comments and Suggestions for Authors

For this round of review, my major concern is this:

Regarding the inclusion of BMI, I would like to emphasize that including it is not only necessary but fundamental to the integrity and completeness of your current research, particularly in the context of a study analyzing weight change. Moreover, I would like to point out that it is a basic principle in research that the same dataset should not be partitioned to create multiple publications when the data are integral to the primary study's conclusions. This practice, often referred to as 'salami slicing', is discouraged as it can lead to fragmented or incomplete reporting of research findings. In your study, the baseline BMI or weight measurements are of immediate importance as they provide crucial context for interpreting any weight changes observed. Without this baseline data, the significance, direction, and magnitude of the weight changes cannot be fully understood or accurately assessed.

Some other comments may include the following:

The current format "BIKTARVY" appears to be missing the appropriate trademark symbol. To align with best practices and legal standards, it is advisable to use the registered trademark symbol (®) after the first mention of "BIKTARVY" in your manuscript. No need for quotation marks.

I recommend revising the section on statistical methodology to enhance clarity, particularly regarding the choice of statistical tests, with specific emphasis on the application of the z-test. It is essential to elaborate on the specific conditions that warranted the use of the z-test over the more commonly employed t-test. An explicit explanation of where and why the z-test was utilized in your analysis is needed, as the rationale for opting for the z-test is not immediately clear. Additionally, addressing the assumption of normality is crucial. Mentioning and describing any normality tests employed (such as the Shapiro-Wilk test, Kolmogorov-Smirnov test, Q-Q plots, etc.) to validate these assumptions is necessary, especially given the distinction you make between the use of the t-test and the z-test in your analysis.

Furthermore, in the methodology section, please elaborate on the rationale behind your choice of age limit for splitting the group for the subsequent analysis. It is important to analyze and define this rationale clearly, especially since no relevant studies support this demarcation.

Now, I would like to draw your attention to the interpretation of the data presented in Table 2, as the accompanying text seems to overgeneralize the findings.

The statement "All of the symptoms presented in the data are observed at a higher percentage in the older age group, and this difference is not merely coincidental" needs reconsideration, particularly in light of the p-values. It is essential to note that not all the differences in symptom prevalence between the age groups reach statistical significance. Please revise the text to reflect a more precise interpretation of the data, acknowledging the symptoms for which the age-related differences are statistically significant and those for which they are not.

In all tables, please avoid the use of “older/younger age group” and rather provide the exact demarcation employed, i.e., > or < 45 years.

Regarding Figure 1 in your manuscript, the current format could potentially be misleading. In longitudinal analysis, it is standard practice to represent time on the x-axis. I recommend employing a box plot or similar graphing technique to effectively display the IQR and median. In addition, if you choose to present these data in a figure, Table 5 becomes automatically redundant.

Author Response

For this round of review, my major concern is this:

Regarding the inclusion of BMI, I would like to emphasize that including it is not only necessary but fundamental to the integrity and completeness of your current research, particularly in the context of a study analyzing weight change. Moreover, I would like to point out that it is a basic principle in research that the same dataset should not be partitioned to create multiple publications when the data are integral to the primary study's conclusions. This practice, often referred to as 'salami slicing', is discouraged as it can lead to fragmented or incomplete reporting of research findings. In your study, the baseline BMI or weight measurements are of immediate importance as they provide crucial context for interpreting any weight changes observed. Without this baseline data, the significance, direction, and magnitude of the weight changes cannot be fully understood or accurately assessed.

Thank you very much for your explanation and suggestions. We will take your comments into consideration for future reference. Also, we have added a table in the article that includes data related to BMI.

Some other comments may include the following:

The current format "BIKTARVY" appears to be missing the appropriate trademark symbol. To align with best practices and legal standards, it is advisable to use the registered trademark symbol (®) after the first mention of "BIKTARVY" in your manuscript. No need for quotation marks.

We have removed the quotation marks and used the registered trademark symbol. Thank you very much for the suggestion.

I recommend revising the section on statistical methodology to enhance clarity, particularly regarding the choice of statistical tests, with specific emphasis on the application of the z-test. It is essential to elaborate on the specific conditions that warranted the use of the z-test over the more commonly employed t-test. An explicit explanation of where and why the z-test was utilized in your analysis is needed, as the rationale for opting for the z-test is not immediately clear. Additionally, addressing the assumption of normality is crucial. Mentioning and describing any normality tests employed (such as the Shapiro-Wilk test, Kolmogorov-Smirnov test, Q-Q plots, etc.) to validate these assumptions is necessary, especially given the distinction you make between the use of the t-test and the z-test in your analysis.

We genuinely appreciate your thoughtful feedback and recommendations regarding the statistical methodology section of the report. Improving the clarity of our methodology is indeed a crucial aspect, and we will definitely take your suggestions into account as we make revisions. We've already gone ahead and revised the section to provide a more detailed explanation of why we chose the z-test over the t-test. This includes presenting clear specifics about the conditions that justified our choice, making it more evident in the analysis. We've also added an explanation regarding where and why the z-test was employed in our analysis. Furthermore, we recognize the importance of addressing the assumption of normality in our statistical analysis. To meet this requirement, we will mention and describe the normality tests we used, such as the Shapiro-Wilk test. These additional pieces of information will enhance transparency and provide assurance regarding the validity of our statistical assumptions.

Furthermore, in the methodology section, please elaborate on the rationale behind your choice of age limit for splitting the group for the subsequent analysis. It is important to analyze and define this rationale clearly, especially since no relevant studies support this demarcation.

In this section, we have described the rationale behind choosing to divide the groups in this manner.

Now, I would like to draw your attention to the interpretation of the data presented in Table 2, as the accompanying text seems to overgeneralize the findings.

The statement "All of the symptoms presented in the data are observed at a higher percentage in the older age group, and this difference is not merely coincidental" needs reconsideration, particularly in light of the p-values. It is essential to note that not all the differences in symptom prevalence between the age groups reach statistical significance. Please revise the text to reflect a more precise interpretation of the data, acknowledging the symptoms for which the age-related differences are statistically significant and those for which they are not.

Thank you for your observation. We have revised the description of this table.

In all tables, please avoid the use of “older/younger age group” and rather provide the exact demarcation employed, i.e., > or < 45 years.

We have revised these terms, thank you for pointing them out.

Regarding Figure 1 in your manuscript, the current format could potentially be misleading. In longitudinal analysis, it is standard practice to represent time on the x-axis. I recommend employing a box plot or similar graphing technique to effectively display the IQR and median. In addition, if you choose to present these data in a figure, Table 5 becomes automatically redundant.

I have redone Figure 2 and represented it as a box plot where I have included all the necessary data. Thank you for the suggestion, and we hope that this time it will be much more eloquent.

Thank you again for your time and your recommendations!

Round 4

Reviewer 3 Report

Comments and Suggestions for Authors

Dear authors, there are still several areas that need further clarification and enhancement:

Citations and Rationale in Introduction:

The statements on treatment extending lives and reducing HIV transmission, and the risks associated with disrupted therapeutic regimens, require support from relevant literature:

·       “Although this treatment is not curative, it can extend patients' lives and reduce HIV transmission.”

·       “A disrupted therapeutic regimen not only endangers the health of the individual patient but also increases the risk of transmitting a potentially resistant strain of the virus to others.”

·       “A weight loss of more than 5 kg in a short period can be alarming.”

The assertion regarding weight loss of more than 5kg needs either a citation or a clear explanation of the clinical rationale behind this specific cutoff.

Presentation of BMI and Weight Changes:

BMI data should be included in Table 1.  In fact, all comparisons between groups’ weights should be discussed before weight change results. This will help establish a baseline.

The relationship between BMI and weight changes, particularly in light of the data presented in Table 5, warrants a more comprehensive critical analysis. A notable contradiction emerges from the results: while weight changes occur at varying rates across different age groups, the overall BMI for these groups remains consistent. While theoretically possible, this observation raises questions about the interpretative value and implications of your findings. Clarifying and discussing this apparent discrepancy is essential for the meaningfulness of the study.

Clarifications and Visualizations:

Please define the specific timepoint for the bar chart comparison in Figure 1. Is it the baseline measurement?

Maintaining consistency in group titles (18-45 Years old/45+ or group1/2) throughout the tables is crucial for clarity and ease of understanding.

Clarify the results' description of Table 3 regarding weight maintenance and the statistical significance. It is unclear what the mean ± standard deviation (SD) represents in Table 3 for the maintenance group. Consider visualizing weight-related results for greater impact.

The current visualization emphasis on CD4 count increase, a well-known effect of antiretroviral therapy (ART), does not add significant insight. Comparing CD4 count increases between age groups might be more informative.

Looking forward to seeing these improvements in your revised manuscript.

Author Response

Dear authors, there are still several areas that need further clarification and enhancement:

Citations and Rationale in Introduction:

The statements on treatment extending lives and reducing HIV transmission, and the risks associated with disrupted therapeutic regimens, require support from relevant literature:

  • “Although this treatment is not curative, it can extend patients' lives and reduce HIV transmission Although this treatment is not curative, it can extend patients' lives and reduce HIV transmission.”

Following your observation, we have added the corresponding number of the article from which this statement was taken, even though it was mentioned one line below.

  • “A disrupted therapeutic regimen not only endangers the health of the individual patient but also increases the risk of transmitting a potentially resistant strain of the virus to others.”

I have added the source of this information.

  • “A weight loss of more than 5 kg in a short period can be alarming.”

Also, the source of this information was cited one line below (number 11), but to avoid any confusion, I have mentioned it immediately after the sentence.

The assertion regarding weight loss of more than 5kg needs either a citation or a clear explanation of the clinical rationale behind this specific cutoff.

In medical practice, a weight loss or gain of around 5 kg is often considered a sensitive and significant indicator of health changes. This amount is large enough to be clinically relevant, yet not so large that it's rare among patient populations. Therefore, 5 kg serves as a useful benchmark for quickly identifying patients who may need further investigation or adjustments in their treatment. Setting a specific threshold of 5 kg for weight changes helps standardize patient monitoring and promptly identifies those who might benefit from nutritional interventions or treatment adjustments. This can be vital for maintaining overall health and long-term quality of life for patients. Additionally, antiretroviral therapy, including 'Biktarvy', can cause gastrointestinal side effects which in turn might affect appetite and nutrient absorption. A 5 kg weight change over a short period can be a sign of such side effects and, therefore, an important marker for assessing the drug's tolerability.

I have added this information to the manuscript as well.

Presentation of BMI and Weight Changes:

BMI data should be included in Table 1.  In fact, all comparisons between groups’ weights should be discussed before weight change results. This will help establish a baseline.

The relationship between BMI and weight changes, particularly in light of the data presented in Table 5, warrants a more comprehensive critical analysis. A notable contradiction emerges from the results: while weight changes occur at varying rates across different age groups, the overall BMI for these groups remains consistent. While theoretically possible, this observation raises questions about the interpretative value and implications of your findings. Clarifying and discussing this apparent discrepancy is essential for the meaningfulness of the study.

With all due respect, we believe that Table 5 cannot be incorporated into Table 1 as it does not fit this table format, having different column headers and presenting information from a different category. However, to accommodate your requested changes, we have moved Table 5 to follow Table 1 (with Table 5 thus becoming Table 2), as we agree that it aids in establishing a baseline. Regarding weight changes, we noted that " It is noteworthy that after 6 months of treatment, even though there were fluctuations in weight, these were not significant enough for patients to be reclassified into a differ-ent weight category. " which means that the BMI has changed, but not enough to shift patients into a different weight category. Rest assured, these results are carefully collected, interpreted, and most importantly, 100% accurate.

Clarifications and Visualizations:

Please define the specific timepoint for the bar chart comparison in Figure 1. Is it the baseline measurement?

The comparison in the chart from Figure 1 was made based on the measurements of the patients taken at the time of their enrollment in the study. We have also mentioned this in the manuscript.

Maintaining consistency in group titles (18-45 Years old/45+ or group1/2) throughout the tables is crucial for clarity and ease of understanding.

Thank you for your observation, we have modified the group titles in all tables so that the same naming is used consistently throughout.

Clarify the results' description of Table 3 regarding weight maintenance and the statistical significance. It is unclear what the mean ± standard deviation (SD) represents in Table 3 for the maintenance group. Consider visualizing weight-related results for greater impact.

As explained in the text, Table 3 (which, following the modifications, has now become Table 4) shows the seven individuals from both groups (2 patients from the first group and 5 patients from group 2) who lost more than 5 kg in weight. The mean and SD in the table represent the average and standard deviation of the number of kilograms lost in weight, which is greater than 5 (Group 1: 5.2 kg, 5.6 kg; Group 2: 5.4 kg, 5.7 kg, 5.5 kg, 8.1 kg, 6.1 kg). We have included this description in the manuscript as well.

The current visualization emphasis on CD4 count increase, a well-known effect of antiretroviral therapy (ART), does not add significant insight. Comparing CD4 count increases between age groups might be more informative.

This is not the subject of the article, but nevertheless, we have added more information regarding this in the manuscript. Thank you for the observation.

Looking forward to seeing these improvements in your revised manuscript.
